# Preparation of Oily Sludge-Derived Activated Carbon and Its Adsorption Performance for Tetracycline Hydrochloride

**DOI:** 10.3390/molecules29040769

**Published:** 2024-02-07

**Authors:** Jie Long, Piwen He, Krzysztof Przystupa, Yudie Wang, Orest Kochan

**Affiliations:** 1School of Urban Construction, Yangtze University, Jingzhou 434023, China; jielong.stu@yangtzeu.edu.cn (J.L.); butterfly.stu@yangtzeu.edu.cn (Y.W.); 2Department of Automation, Lublin University of Technology, 20-618 Lublin, Poland; 3School of Computer Science, Hubei University of Technology, Wuhan 430068, China; orest.v.kochan@lpnu.ua; 4Department of Measuring-Information Technologies, Lviv Polytechnic National University, Bandery Str. 12, 79013 Lviv, Ukraine

**Keywords:** oily sludge, activated carbon, tetracycline hydrochloride, KOH activation, adsorption

## Abstract

Oily sludge-derived activated carbon was prepared using the potassium hydroxide (KOH) activation method using oily sludge as a raw material, and one-factor experiments determined the best conditions for preparing activated carbon. The activated carbon’s morphological structure and surface chemical properties were analyzed by scanning different characterization tools, and the adsorption behavior of tetracycline hydrochloride was investigated. The results showed that the optimum conditions for preparing oily sludge-derived activated carbon were an activation temperature of 400 °C, activation time of 30 min, activator concentration of 1 mol/L, and impregnation ratio of 2 mL/g. After activation, the activated carbon had more pores and a more orderly crystal structure arrangement, the specific surface area was 2.07 times higher than that before activation, and the surface was rich in functional groups such as -HO, -C-O, -C=C, and -C-H, which increased the active sites of activated carbon. Physicochemical effects dominated the adsorption process. It belonged to the spontaneous heat absorption process under the quasi-secondary kinetic and Langmuir isothermal models. The maximum monolayer adsorption capacity of KOH-activated carbon was 205.1 mg·g^−1^.

## 1. Introduction

TCH (tetracycline hydrochloride) is a common antibiotic, and most of the tetracyclines are biologically metabolized, excreted and accumulate in large quantities in the natural environment. It has been shown that antibiotics affect the structure and enzymatic activity of soil bacterial colonies, as well as the development of antibiotic resistance, which is related to antibiotics’ effectiveness in treating human diseases. Therefore, it is necessary to find effective measures to remove TCH.

There are various treatment methods for tetracycline, including electrochemical methods [1], advanced oxidation [2], ion exchange [3], membrane methods [4], photocatalysis [5,6], and adsorption [7]. Among them, the advanced oxidation method makes it challenging to synthesize catalysts. The electrochemical and membrane methods are more costly, and the treatment process is more complicated. Therefore, the adsorption method has received wide attention because of its convenient operation, low cost, and no secondary pollution. Activated carbon, resin, graphene, and carbon nanotubes have been used as adsorbent materials, among which activated carbon has received widespread attention due to its vast source of raw materials and low cost. Activated carbon is a solid rich in carbon obtained by pyrolysis of biomass in an anoxic environment. It is considered an ideal adsorbent due to its large specific surface area, numerous pore structures, abundant functional groups, and renewable resources [8]. Its preparation process mainly includes carbonization and activation. Carbonization aims to obtain an initial pore structure and a carbon concentrate with appropriate mechanical strength for activation. The purpose of activation is to create new pores and expand the specific surface area of activated carbon [9], sludge [10] and rice husk [11], and wood chips [12] are often used as raw materials for the preparation of activated carbon. The residue after the pyrolysis of oily sludge has the advantage of higher carbon content and is an excellent raw material for preparing activated carbon.

Oily sludge is a type of polluted waste created during the exploration, extraction, cleaning, refining, storage, and transport of crude oil. It is also produced due to accidents, natural sedimentation, and other factors [13]. Its primary constituents are water (30–50%), oil (30–80%), and solids (10–20%). The oily component contains heavier oil, asphaltene, and more saturated hydrocarbons. China produces more than 5 million tonnes of oily sludge annually, classified as hazardous waste HW08 on the National Hazardous Waste List [14]. Currently, the primary methods for treating oily sludge are simple landfilling and advised incineration, which result in resource waste and environmental pollution. To find a new use for oily sludge that is low-polluting, inexpensive, high value-added, and can meet the standards of “three-chemistry” waste treatment, it is one of the research hotspots for academics both at home and abroad. The pore size and specific surface area limit the adsorption performance of the original activated carbon. The activation energy increases the activated carbon’s specific surface area and pore volume and enriches the pore structure. The activation methods are divided into chemical and physical activation; chemical modification includes acid and alkali modification, oxidizer modification, and metal modification, and physical activation includes ball milling and gas blowing. The chemical activation method has the advantages of low activation temperature, short activation time, high specific surface area and controllable pore structure. Thus, it is widely adopted [15]. ZnCl_2_ and other activators have problems with corrosion and low chemical recovery. Using KOH as the activator can effectively improve the adsorption capacity of activated carbon for a variety of pollutants; increase the content of hydroxyl groups, carboxyl groups and other functional groups in the biochar; improve the pore structure of the biochar; and change the adsorption performance [16]. Liu [17] and Han [18] et al. have investigated the adsorption of organic materials and heavy metals on oily sludge-derived activated carbon.

Nevertheless, tetracycline hydrochloride has been observed to adsorb on activated carbon very infrequently. Based on the above considerations, the activated carbon was prepared from oil-containing sludge using KOH as the activator in this paper. The adsorption effect of activated carbon on TCH was selected as the evaluation index, and a one-factor experiment optimized the preparation of activated carbon. Various instruments were used to study the morphological structure and surface chemical properties of the activated carbon and to investigate the factors affecting the adsorption of TCH, and isothermal adsorption, kinetic and thermodynamic models were used to investigate the adsorption process, which provided a new idea for the treatment of wastes with wastes.

## 2. Results and Discussion

### 2.1. Optimization of Preparation Conditions

#### 2.1.1. Activation Temperature

Under the conditions of a KOH concentration of 3 mol/L, impregnation ratio of 2 mL/g, and activation time of 90 min, the effects of activation temperature (200–500 °C) on the adsorption of TCH by OSAC-KOH were examined. The results are shown in Figure 1a.

As shown in Figure 1a, the adsorption of TCH by OSAC-KOH showed a tendency to increase and then decrease with the increase in activation temperature, and reached the maximum value at 400 °C. This tendency may be due to the reaction of KOH at high temperatures of Formula (1):(1)$$6KOH+2C\to 2K+2K_{2}CO_{3}+3H_{2}$$

Less gas is produced when the activation temperature is 200 °C because the reaction between KOH and C and its organic content is incomplete. The gas yield increased as the activation temperature was raised to 400 °C. The diffusion of gas and embedding of K monomers increased the number of OSAC micropores [19], which in turn increased the adsorption of TCH by OSAC-KOH from 68.8 mg·g^−1^ to 96.1 mg·g^−1^. As the temperature increased, the adsorption amount gradually reduced to 47.7 mg·g^−1^. This is because high temperatures induce the microporous pores in OSAC to collapse; as a result, there are fewer micropores and more mesopores and macropores, thus reducing the activated carbon’s ability to adsorb substances. Therefore, the best activation temperature to produce OSAC-KOH was 400 °C.

#### 2.1.2. Activation Time

Under an impregnation ratio of 2 mL/g, an activation temperature of 400 °C, and an activator concentration of 3 mol/L, the effects of activation time (10–120 min) on the adsorption of TCH by OSAC-KOH were examined. The results are shown in Figure 1b.

The adsorption of TCH by OSAC-KOH showed a growing and declining trend as activation time increased, as shown in Figure 1b, reaching the maximum value of 55.5 mg·g^−1^ at 30 min. At a 10 min activation time, OSAC activation was not complete; however, a longer activation time would result in sintering and the collapse of the pores [20], and damage to the produced pores from overloading the activator with OSAC [21]. The carbon skeleton was also damaged to a certain extent. The carbon atoms on it were consumed, developing the original micropores into medium and large pores. The adsorption amount of OSAC-KOH was thus reduced, which agreed with Nasrullah’s [22] study. Therefore, the best activation time to produce OSAC-KOH was 30 min.

#### 2.1.3. Activator Concentration

Under the conditions of the impregnation ratio of 2 mL/g, the activation temperature of 400 °C, and the activation time of 30 min, the effects of KOH concentration (0.3–2 mol/L) on adsorption of TCH by OSAC-KOH were examined. The results are shown in Figure 1c.

As shown in Figure 1c, the adsorption of TCH by OSAC-KOH was maximized at a KOH concentration of 1 mol/L, which was 92.2 mg·g^−1^. The activation level of OSAC was low, and pore formation was incomplete when the KOH concentration was less than 1 mol/L, which impacted its adsorption effect. The amount of adsorption reduced when the concentration reached 2 mol/L. This could result from the increased concentration of KOH adhering to the OSAC’s surface, which hinders the release of the gas produced inside and decreases the size of the micropores. In addition, at a particular impregnation ratio, the increase in KOH concentration blocked portions of the OSAC’s micropores, which were not entirely unblocked by acid and water washing, lowering the material’s adsorption capacity. Therefore, the best activator concentration to produce OSAC-KOH was 1 mol/L.

#### 2.1.4. Impregnation Ratio

Under a temperature of 400 °C, for 30 min, and with a KOH concentration of 1 mol/L, the effects of the impregnation ratio (1–4 mL/g) on the adsorption of TCH by OSAC-KOH were examined. The results are shown in Figure 1d.

One of the primary variables influencing the porosity of activated carbon is the impregnation ratio [23]. As shown in Figure 1d, the adsorption of TCH by OSAC-KOH showed a tendency to increase and then decrease with the increase in the impregnation ratio, and reached the maximum at 2 mL/g, which was 77.0 mg·g^−1^. Sait Yorgun [24] showed that the rise in the impregnation ratio significantly affected the specific surface area, microporous surface area, and volume of the activated carbon at an activation temperature of 400 °C. The increase in potassium hydroxide concentration might prevent the tar from clogging the pore micropores when the impregnation ratio was less than 2 mL/g, enhancing the adsorption action of activated carbon. When the impregnation ratio was raised to 4 mL/g, the extra potassium hydroxide either entered the OSAC’s original pores or caused the pore wall to vaporize, creating macropores between adjacent pores. As a result, the specific surface area of OSAC-KOH shrank, and its ability to adsorb TCH declined [25]. Therefore, the best impregnation ratio to produce OSAC-KOH was 2 mL/g.

### 2.2. Characterization of Activated Carbon

Activated carbon was subjected to an SEM analysis both before and after activation. From Figure 2a, it can be seen that the OSAC surface is smooth, the number of pores is small, and the pore size is large. According to Figure 2b, the surface of OSAC-KOH that has been modified by KOH is rough and uneven, has many micropores, and has a rich pore structure. In Figure 2c, the EDS characterization tests detect the presence of K; its mass fraction, which is 18.84%; and KOH load success on the surface of activated carbon. C and O are the main elements, indicating that KOH activation can effectively form a layered structure, improve the porosity of OSA-KOH, and give OSA-KOH a larger specific surface area.

Compared to the surface before activation, the specific surface area has increased, which is helpful for TCH adsorption. FTIR and XRD techniques were used to examine the activated carbon’s crystal structure and surface functional groups before and after activation. From Figure 3a, it can be seen that the more substantial absorption peaks, corresponding to (002) crystal surface [14], arose around 2θ = 26° both before and after activation by activated carbon. The surface pores of the activated carbon extend into the interior, according to the narrow and weak diffraction peaks associated with diffraction angles of about 20.5, 27, 49.5, and 59.5°, respectively. After activation, the diffraction peak intensities increased, indicating a more ordered crystal structure for OSAC-KOH. According to Figure 3b, the stretching vibrations of the -HO, -C-O, -C=C, and -C-H bonds are represented by the absorption peaks at 3401, 1697, and 1384, 779 cm^−1^, respectively. After KOH activation, the strength of the absorption peaks at -HO, -C-O, -C=C, and -C- increased, and the addition of more of these functional groups may have increased the adsorption of TCH through higher electrostatic attraction, hydrogen bonding, and π-π.

The elemental compositions of OSAC and OSAC-KOH were analyzed by XPS, as shown in Figure 4. The content of O element in activated carbon increased after KOH modification; this is due to the doping of O from the activator into the activated carbon [26]. C1s peak binding energy with 284.6 eV as the center was used as the calibration standard, and the C peak is convoluted into four characteristic peaks: 284.6 eV(C-C/C-H), 285.8 eV(C-O), 287.0 eV(C=O), and 289.5 eV(O=C-O) [27]. The results showed that KOH activation reduced the atomic ratio of C-C/C-H groups. In the O1s spectrum, the 531.95 eV and 533.42 eV peaks were assigned to C=O and C-O, respectively. After KOH modification, C−O replaces C=O as the leading O-containing group. KOH modification can significantly affect the surface functional groups of biochar, especially the oxygen-containing groups [28].

Figure 5a,b show the N_2_ adsorption–desorption isotherms and pore size distribution of activated carbon before and after activation, respectively. As shown in Figure 5a, the adsorption and desorption of nitrogen by OSAC and OSAC-KOH conformed to the type IV isotherm according to the classification criteria defined by IUPC. At p/p_0_ > 0.4, the nitrogen adsorption of both activated carbons increased, indicating that both OSAC and OSAC-KOH have microporous and mesoporous structures, and OSAC-KOH has a higher microporous and mesoporous content. As shown in Figure 5b and Table 1, the specific surface area, total pore volume, and average pore diameter of OSAC-KOH after KOH activation were 10.1938 m^2^·g^−1^, 0.0211 cm^3^·g^−1^ and 20.9481 nm, respectively, which were 2.07-, 2.22-, and 2.0-fold higher compared to that of OSAC, and thus possessed more adsorption sites and a more robust adsorption performance.

### 2.3. Adsorption Influences

#### 2.3.1. Liquid pH

The main factor influencing the adsorption process is the pH of the solution [29]. As shown in Figure 6a, the adsorption of TCH by OSAC-KOH increased from 34.0 mg·g^−1^ to 107.0 mg·g^−1^ as the solution pH was increased from 3 to 4. Continuing to increase the pH to 11, the adsorption gradually decreased to 24.0 mg·g^−1^. This may be because activated carbon is amphoteric, and its capacity to absorb cationic or anionic chemicals is influenced by the characteristics of its surface [30]. The TCH solution was dominated by H_3_L^+^ at pH < 3.4, H_2_L at 3.4 < pH < 7.6, HL^−^ at 7.6 < pH < 9.0, and L^2−^ at 9.0 < pH. At a low pH, OSAC-KOH’s surface had a higher positive charge, which made H_3_L^+^ electrostatically reject it. When the solution was alkaline, TCH in the solution mainly existed in anionic form, and the oxidized groups on the surface of OSAC-KOH were gradually ionized. The surface charge density and adsorption decreased with the increase in pH, which could be attributed to the electrostatic repulsion between the TCH and the reactive chemical groups of OSAC-KOH decreasing the adsorption of TCH on OSAC-KOH. Additionally, under alkaline circumstances, some active carboxyl sites on the surface of the activated carbon would be passivated or even occluded [31], which reduced the OSAC-KOH’s ability to adsorb. This suggests that electrostatic attraction plays an important role in the adsorption process for TCH.

#### 2.3.2. TCH Initial Concentration

As shown in Figure 6b, as the initial concentration increased, the amount of TCH adsorbed by OSAC-KOH rose from 36.6 mg·g^−1^ to 119.6 mg·g^−1^. This fact is primarily caused by a high concentration of TCH solution, which offers more mass transfer driving forces and adsorption sites and increases the possibility that OSAC-KOH and TCH molecules would collide, increasing its adsorption capacity. When the initial concentration was higher than 350.0 mg/L, the OSAC-KOH adsorption site was saturated, and the adsorption amount was, thus, brought to equilibrium.

#### 2.3.3. Adsorption Temperature and Time

As shown in Figure 6c, the adsorption of TCH by OSAC-KOH increased from 107.0 mg·g^−1^ to 131.0 mg·g^−1^ when the temperature was increased from 25 to 45 °C. This is because warming reduces the resistance to movement, which accelerates the diffusion of TCH molecules [32]. At the same time, the activation energy of the OSAC-KOH active adsorption site becomes higher, the adsorption site increases [33], and its adsorption capacity is thus increased. As shown in Figure 5d, the amount of TCH adsorbing onto OSAC-KOH grew initially, slowed down with time, and ultimately stabilized. OSAC-KOH possessed a lot of adsorption sites during the first 360 min of adsorption, and its adsorption capacity reached 104.0 mg·g^−1^. As the adsorption time continued to increase, the internal adsorption sites were occupied by TCH and saturated, and the adsorption amount was stabilized to 105.0 mg·g^−1^.

### 2.4. Kinetic Model

#### 2.4.1. Adsorption Kinetics

The following kinetic equations were used to fit the experimental results using the quasi-primary kinetic model (2) and the quasi-secondary kinetic model (3):(2)$$q_{t}=q_{e}(1-e^{k_{1}t})$$
(3)$$q_{t}=\frac{k_{2}{q_{e}}^{2}t}{1+k_{2}q_{e}t}$$
herein, q_e_, q_t_—OSAC-KOH adsorption and adsorption equilibrium at time t, mg·g^−1^; k_1_, k_2_—kinetic adsorption rate constants for the quasi-primary and quasi-secondary states; t—absorption time, min.

Adsorption kinetics primarily focus on modelling the adsorption rate to determine whether physisorption or chemisorption is the adsorption process. As shown in Figure 7 and Table 2, the equilibrium adsorption of TCH by OSAC-KOH obtained from the quasi-primary and quasi-secondary kinetic fits were 104.2 mg·g^−1^ and 118.9 mg·g^−1^, respectively. The latter quasi-secondary kinetic R_1_^2^ (0.9951) was also higher than the quasi-secondary kinetic R_2_^2^ (0.9892), which suggests that the process is an adsorption process dominated by chemisorption [34], with a combination of physical and chemical adsorption. It agrees with the characterization results, indicating that OSAC-KOH has a rich pore structure enriched with functional groups such as -HO, -C-O, -C=C, and -C-H.

#### 2.4.2. Isothermal Adsorption

The following isothermal equations were used to match the experiments using two isothermal models: Langmuir (Equation (4)) and Freundlich (Equation (5)):(4)$$q_{e}=\frac{q_{m}c_{e}b}{1+bc_{e}}$$
(5)$$q_{e}=k{c_{e}}^{\frac{1}{n}}$$
herein, b, k—adsorption equilibrium constants in the Langmuir and Freundlich model; c_e_—TCH concentration at equilibrium adsorption, mg/L; q_e_—OSAC-KOH adsorption at equilibrium adsorption, mg·g^−1^; q_m_—TCH adsorption is greatest on the monolayer of OSAC-KOH’s surface, mg·g^−1^; n—adsorption density constant.

As shown in Figure 8 and Table 3, the Langmuir isothermal model R^2^ of both OSAC-KOH and OSAC was more significant than that of the Freundlich isothermal model R^2^, and the former had a higher correlation with the fitting parameters of the experimental data, which indicated that the adsorption of TCH by OSAC-KOH was more consistent with the Langmuir model. The distribution of adsorption sites is homogeneous and constrained in both adsorption processes, proving that they are both monolayer adsorptions. OSAC-KOH had a monolayer adsorption capacity that was 2.9 times more than OSAC at 105.1 mg·g^−1^. The Langmuir model equation with 1/n(0.3346) < 1 also indicates that the adsorption of TCH on OSAC-KOH is a more likely process [35]. Table 4 compares the greatest amount of TCH that can be absorbed by activated carbon made from various raw materials, and it is clear that OSAC-KOH exhibits strong adsorption capabilities.

#### 2.4.3. Adsorption Thermodynamics

The adsorption thermodynamic parameters Gibbs free energy (ΔG^θ^), enthalpy change (ΔH^θ^) and entropy change (ΔS^θ^) were calculated using Equations (6) and (7) with the following equations.
(6)$$\Delta G=-RT\ln K^{\theta }$$
(7)$$\ln K^{\theta }=\frac{\Delta S^{\theta }}{R}-\frac{\Delta H^{\theta }}{RT}$$
where R—gas constant, 8.314 J·(mol·K)^−1^; T—absolute temperature K; K^θ^—factor of equilibrium distribution.

Table 5 demonstrates that ΔG^θ^ is smaller than 0 at various temperatures, indicating that adsorption is spontaneous. Its value also lowers as the temperature rises, suggesting that high temperatures aid adsorption. ΔH^θ^ > 0, which is 17.19 kJ·mol^−1^, indicates that the adsorption of TCH on OSAC-KOH is a heat-absorbing reaction. ΔS^θ^, meaning the degree of disorder at the solid–liquid interface during adsorption, has a value of 66.34 J·(mol·K)^−1^, indicating that the process is becoming more random and disorderly. The adsorption process is consistent with monolayer adsorption [44].

## 3. Materials and Methods

### 3.1. Materials

A certain amount of oily sludge was put into a vacuum pyrolysis furnace, heated to 700 °C with nitrogen as the protective gas at the heating rate of 10 °C/min, and kept at this temperature for 60 min. After the pyrolysis furnace was cooled to room temperature, it was removed and dried, and the primary activated carbon (OSAC) was obtained by grinding it until it passed through a 50-mesh sieve. KOH was used as the activator, and 10 g of OSAC was weighed, mixed with various KOH concentrations (0.3–2 mol/L) at different impregnation ratios (1–4 mL/g), and then placed in a magnetic stirrer for 8 h at 200 rpm. The mixture was then baked until it reached a steady weight. The dried samples were placed in a vacuum pyrolysis furnace, where they were gradually heated to the required temperature (200–700 °C) under a nitrogen atmosphere at a rate of 10 °C/min with varying activation durations (30–120 min). The samples were allowed to cool naturally to room temperature before being immersed in 0.1 mol/L hydrochloric acid for 2 h to remove impurities. After this, the samples were neutralized by rinsing them with ultrapure water, drying them, and passing them through a 50-mesh sieve to obtain oily sludge-derived activated carbon (OSAC-KOH).

### 3.2. Analytical Methods

Oily sludge is from the landing sludge of a treatment plant in Liaohe Oilfield, China. All reagents used in this work were of analytical grade, and all solutions were prepared with double-deionized water. NaOH was purchased from Sinopharm Chemical Reagent Co., Shanghai, China. TCH was acquired from Shanghai yuanye Bio-Technology Co., Ltd., Shanghai, China. HCl was purchased from Chengdu Chron Chemicals Co., LTD., Chengdu, China. The surface morphology of activated carbon that has been coated with gold was examined using scanning electron microscopy (SEM, Tescan MIRA LMS, Brno, Czech Republic). An X-ray diffractometer (XRD, SmartLab SE, Tokyo, Japan) was used to investigate the crystal structure using Cu-K rays at 40 kV and 40 mA. With the help of a Fourier Transform Infrared Spectrometer (FTIR; Thermo Scientific Nicolet 6700, Waltham, MA, USA), surface functional group alterations were examined. Samples were compressed using the KBr technique, and the scanning wavelengths ranged from 400 to 500 cm^−1^. A fully automated specific surface and porosity analyzer (BET, Micromeritics ASAP 2460, Freeport, IL, USA) was used to calculate the activated carbon’s specific surface area and pore size distribution.

### 3.3. Adsorption Experiments

#### 3.3.1. Adsorption Influence Factor Experiment

In total, 100 mL of a particular concentration of TCH solution was added to a conical flask containing 100 mg of OSAC-KOH, and the mixture was agitated for 8 h at 25 °C at 200 rpm. The starting concentration of TCH, c_0_ (50–550 mg/L), the solution pH (3–11), adsorption period t (0–960 min), adsorption temperature T (25–45 °C), and all of these variables were under control. Following filtration via a 0.45 nm membrane, the TCH concentration was measured using liquid chromatography, and Equation (8) was used to determine the equilibrium adsorption.
(8)$$q_{e}=\frac{(c_{o}-c_{e})\times V}{m}$$


#### 3.3.2. Adsorption Kinetic Experiments

In total, 100 mg of OSAC and 100 mg of OSAC-KOH were weighed into separate glass vials, along with 100 mL of TCH solution with a c_0_ of 150 mg/L. The solution’s pH was then changed to 4.0 and agitated at 200 revolutions per minute for 8 h at 25 °C. Following the passage of the samples through a 0.45 m filter membrane, samples were obtained at the predetermined time.

#### 3.3.3. Isothermal Adsorption Experiment

In total, 100 mg of OSAC and 100 mg of OSAC-KOH were added into separate glass vials. In total, 100 mL of TCH solution with a c_0_ of 50–200 mg/L was added. The samples were stirred for 8 h at a rate of 200 rpm per minute at 25, 35, and 45 °C to ascertain the TCH concentration. Each of the above experiments was carried out three times.

## 4. Conclusions

OSAC-KOH produced from oily sludge demonstrated effective TCH adsorption. OSAC-KOH should be prepared under ideal conditions: an activation temperature of 400 °C, activation period of 0.5 h, activator concentration of 1 mol/L, and impregnation ratio of 2 mL/g. The KOH-activated OSAC-KOH micropores were created with more pores, a more complex pore structure, an ordered crystal structure, a more extensive specific surface area, and surfaces that were rich in functional groups, including -HO, -C-O, -C=C, and -C-H. The performance of OSAC-KOH during adsorption was affected by the pH of the solution, starting concentration, adsorption time, and adsorption temperature, with pH having the most significant impact. The maximal adsorption capacity of the monolayer is 105.1 mg·g^−1^, and the adsorption process is a heat-absorption response consistent with the Langmuir isothermal model and the quasi-secondary kinetic model. The monolayer’s highest adsorption capacity was 105.1 mg·g^−1^. The adsorption mechanisms are mainly electrostatic, hydrogen bonding, and π-π interactions.

In our future research, we will study the zeta potential of pH and the adsorption mechanism versus pH.

## Figures and Tables

**Figure 1 molecules-29-00769-f001:**
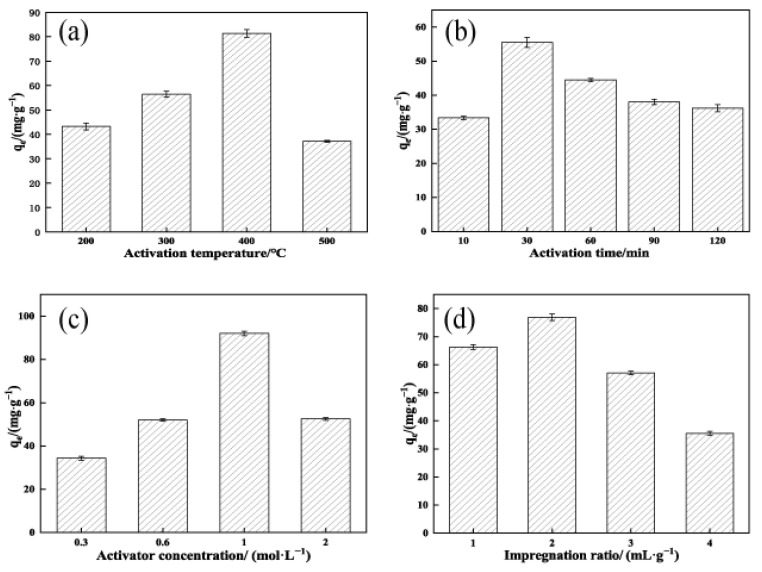
Factors influencing the preparation of OSAC-KOH. (**a**) Activation temperature; (**b**) Activation time; (**c**) Activator concentration; (**d**) Impregnation ratio.

**Figure 2 molecules-29-00769-f002:**
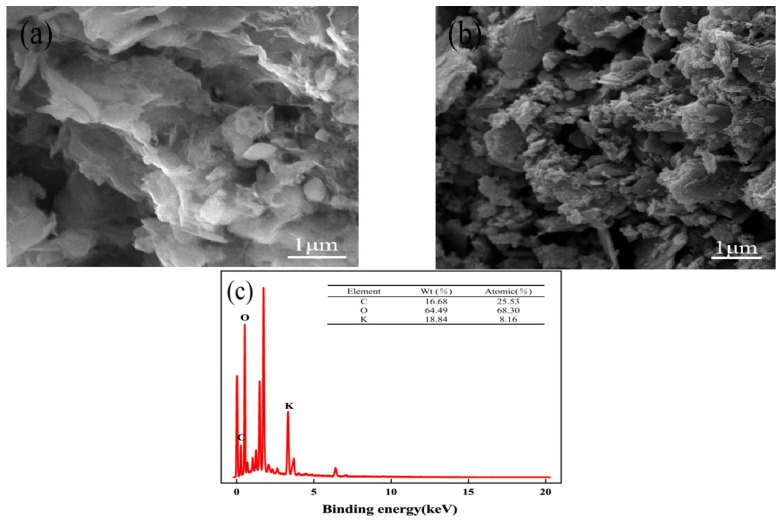
SEM images of activated carbon of (**a**) OSAC, (**b**) OSAC-KOH, and EDS of (**c**) OSAC-KOH.

**Figure 3 molecules-29-00769-f003:**
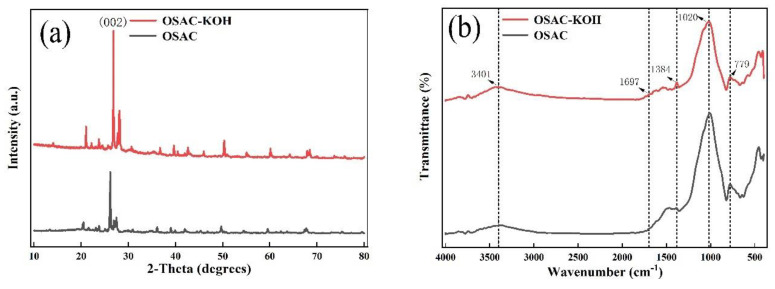
(**a**) XRD patterns; (**b**) FTIR spectra of activated carbons.

**Figure 4 molecules-29-00769-f004:**
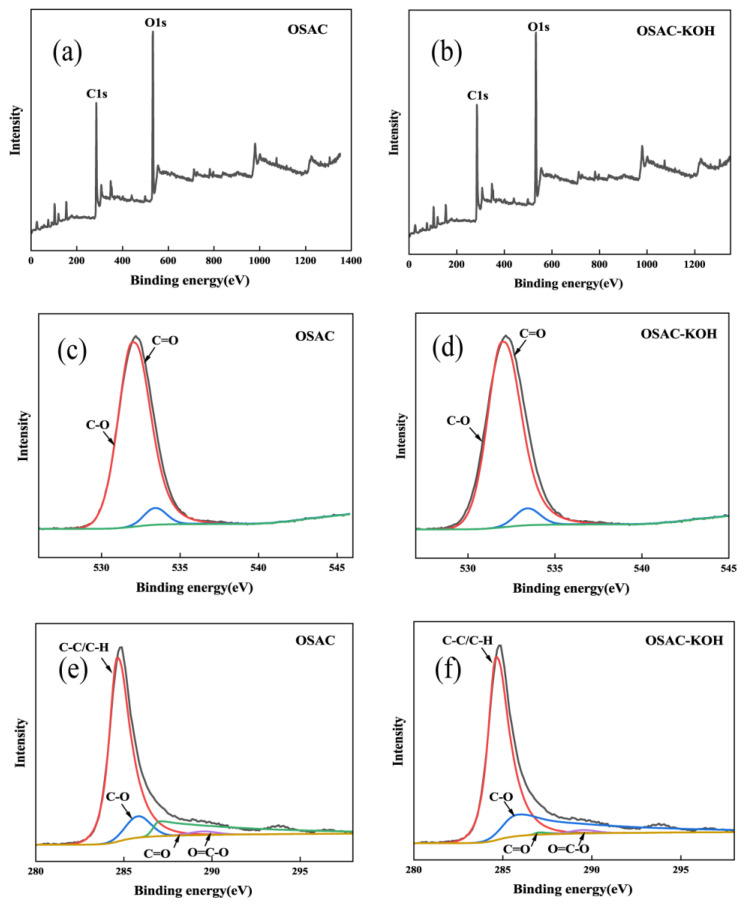
XPS survey (**a**) and (**b**), O1s (**c**) and (**d**), C1s (**e**) and (**f**) of biochars.

**Figure 5 molecules-29-00769-f005:**
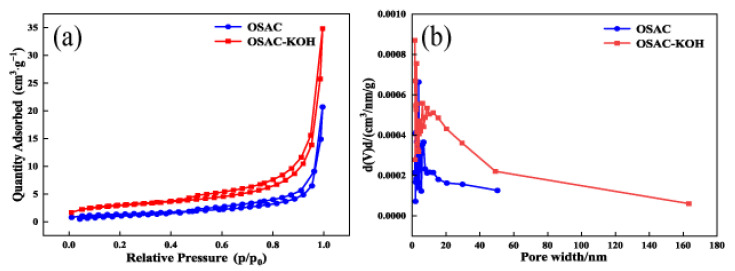
(**a**) N_2_ adsorption–desorption isotherms, (**b**) pore size distribution of activated carbons.

**Figure 6 molecules-29-00769-f006:**
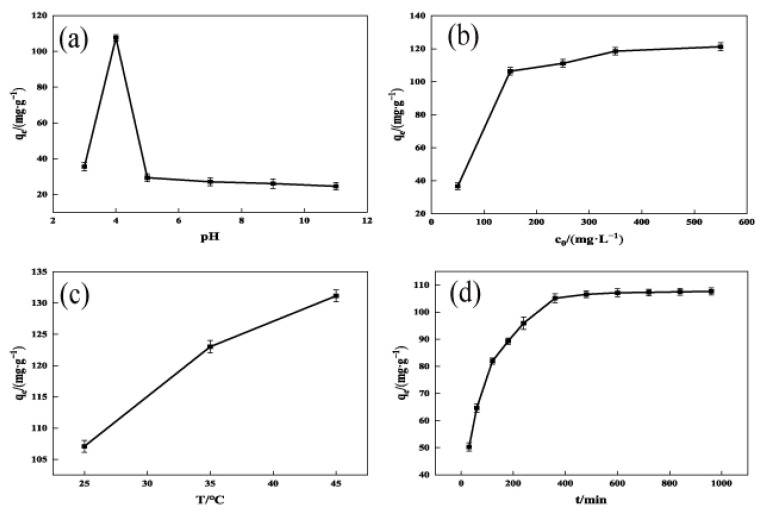
Influencing factors of OSAC-KOH adsorption. (**a**) pH; (**b**) Initial concentration; (**c**) Adsorption temperature; (**d**) Adsorption time.

**Figure 7 molecules-29-00769-f007:**
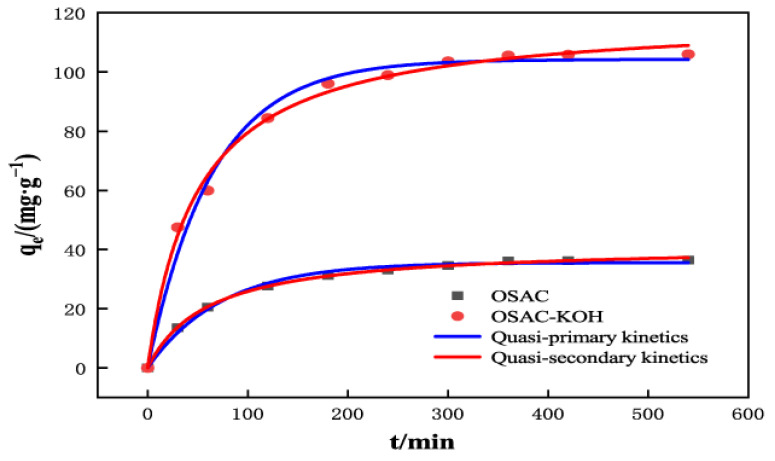
The quasi-first-order kinetic model and quasi-second-order kinetic model of TCH adsorption by OSAC-KOH.

**Figure 8 molecules-29-00769-f008:**
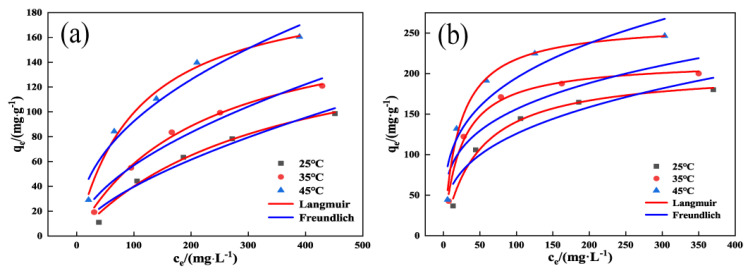
Langmuir isotherm model and Freundlich isotherm model: (**a**) OSAC, (**b**) OSAC-KOH.

**Table 1 molecules-29-00769-t001:** Aperture parameters of OSAC-KOH and OSAC.

Sample	SBET/m^2^·g^−1^	VT/cm^3^·g^−1^	Pore Diameter/nm
OSAC-KOH	10.1938	0.0211	20.9481
OSAC	4.8244	0.0095	9.5031

**Table 2 molecules-29-00769-t002:** Kinetic model fitting parameters of OSAC-KOH and OSAC.

Sample	Quasi-Primary Kinetics			Quasi-Secondary Kinetics		
	q_e_/ (mg·g^−1^)	k_1_/ min^−1^	R_1_^2^	q_e_/ (mg·g^−1^)	k_2_/ (g·mg^−1^·min^−1^)	R_2_^2^
OSAC-KOH	104.21	0.0154	0.9892	118.98	1.69 × 10^−4^	0.9951
OSAC	35.59	0.0135	0.9932	41.43	4 × 10^−1^	0.9991

**Table 3 molecules-29-00769-t003:** Fitting parameters of the adsorption isotherm model of OSAC-KOH and OSAC.

Sample	T/°C	Langmuir			Freundlich		
		q_m_/ (mg·g^−1^)	b/ (L·mg^−1^)	R^2^	K	1/n	R^2^
OSAC-KOH	25	205.11	0.0217	0.9907	26.9365	0.3346	0.8909
	35	216.12	0.0432	0.9928	45.0116	0.2862	0.8574
	45	263.20	0.0477	0.9903	52.0806	0.2700	0.8800
OSAC	25	173.53	0.0029	0.9862	2.2419	0.6335	0.9591
	35	183.95	0.0046	0.9952	4.4942	0.5512	0.9620
	45	205.37	0.0093	0.9900	11.7670	0.4474	0.9471

**Table 4 molecules-29-00769-t004:** Comparison of the maximum adsorption capacity of TCH by activated carbon prepared from different raw materials.

Absorbent	q_m_/(mg·g^−1^)	References
Sludge	4.61	[10]
Bamboo charcoal	22.7	[36]
Zeolite	20.4	[37]
Biomass	58.8	[38]
Rice husk ash	8.37	[39]
Ball-milled biochar	84.5	[40]
Marine sediments	50.0	[41]
Palygorskite	93.3	[42]
Chitosan	13.3	[43]
Oily sludge	126.1	This study

**Table 5 molecules-29-00769-t005:** Adsorption thermodynamic parameter of OSAC-KOH.

Sample	T/K	ΔG^θ^/ kJ·mol^−1^	ΔH^θ^/ kJ·mol^−1^	ΔS^θ^/ J·(mol·K)^−1^
OSAC-KOH	298	−2.173		
	308	−3.802	17.19	66.34
	318	−5.267		

## Data Availability

The data presented in this study are available upon request from the corresponding author. Data cannot be made publicly available due to confidentiality agreements associated with the data.

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
