# Peer review of "Preparation of Oily Sludge-Derived Activated Carbon and Its Adsorption Performance for Tetracycline Hydrochloride"

_molecules, 2024, doi:10.3390/molecules29040769_

Round 1

Reviewer 1 Report

Comments and Suggestions for Authors

Preparation of oily sludge‐derived activated carbon and its adsorption performance for tetracycline hydrochloride

This article studies the optimal conditions for preparing activated carbon from oil sludge as raw material. Through single factor experiments, the optimal preparation conditions were determined. The study on the adsorption behavior of tetracycline hydrochloride shows that the adsorption process is mainly influenced by physical and chemical reactions, which conforms to the quasi second order kinetic model and Langmuir isotherm model. However, major revisions are required before it could be considered for publication. The suggestions are given below.

1.     What are the reasons for choosing KOH activation in the article, and what are the advantages of KOH activation compared to other activators?

2.     Units such as ℃, %, etc. require spaces between numbers. Additionally, the writing of units should be standardized.

3.     The source of the raw materials used in the experiment was not clearly explained and needs to be indicated.

4.     The size and unit writing of the group diagram in Figure 1 are different and need to be adjusted.

5.     The characterization of activated carbon is too simplistic and should be appropriately cited for argumentation and explanation.

6.     XPS is suggested to be added to reveal the heteroatom content of different samples and their influence on the adsorption capacity. Please refer to Journal of Colloid and Interface Science 2024, 653, 1526-1538.

7.     The drawing of the error bar in Figure 5 needs to be clearer.

8.     The article did not plot the zeta potential of pH and did not delve into the adsorption mechanism of pH influence.

9.     The explanation of the adsorption mechanism in the article is relatively simple and not deep enough. It is recommended to read more literatures to provide a detailed explanation of the mechanism.

10.   The references are relatively old, more recently published articles are suggested to be cited, e.g. Coordination Chemistry Reviews 2024, 502, 215612; Journal of Colloid and Interface Science 2022, 607, 933-941.

Comments on the Quality of English Language

Moderate editing of English language is required.

Author Response

The response to the reviews is included in the attached file.

Reviewer 2 Report

Comments and Suggestions for Authors

This paper deals with preparation and characterization of activated carbon made from oily sludge.

Useful results are obtained in this study.

This study will be worthy to be published on this journal after some corrections.

Some suggestions are as follows.

Comment 1: unit

The units should be unified.

You use “mL” in many places and “ml” in p.4, line 182.

Comment 2: Figure 4 horizontal axis

The horizontal axis starts from “-20” in Fig. 4 b). The value cannot be minus. It should be starts from ”0”.

Comment 3: add “(a)” and “(b)”

The indication “(a)” and “(b)” should be added in Figures 2 and 4.

In addition, “a)” and “b)” in the figure caption of Figures 3 and 4 should be altered to “(a)” and “(b)”.

Comment 4: superscript and subscript

You use “N2”, “cm-1”, “m2g-1”, “cm3g-1” and so on in this paper. Can you use superscript and subscript?

Comment 5: expression of units

You use “mg/g” and “mg∙g” in this paper. The slashes and points are mixed. The expression of units should be unified.

Author Response

(The authors gave the same response as above.)

Reviewer 3 Report

Comments and Suggestions for Authors

In this work, Long et al. prepared OSAC‐KOH from oily sludge with effective tetracycline hydrochloride (TCH) adsorption. The morphological structure and surface chemical properties of activated carbon were analyzed. After activation, the activated carbon has more pores and a more orderly crystal structure arrangement before activation. The adsorption process was dominated by physicochemical effects and belonged to the spontaneous heat absorption process, which was under the quasi‐secondary kinetic model and Langmuir isothermal model. Overall, the current work can be accepted by Molecules after several minor revisions, as below:

1) The abstract can be simple. For example, the authors stated 'After activation, the activated carbon has more pores and a more orderly crystal structure arrangement, the specific surface area is 2.07 times higher than that before activation, and the surface is rich in functional groups such as ‐HO, ‐C‐O, ‐C=C, and ‐C‐H.' What is the role of functional groups in this work? It looks like no related information in the Abstract. Similar sentences can be improved. 

2) Introduction. The supporting reference is not enough in the Introduction part. For example, line 45-46, Methods commonly used to remove antibiotics include adsorption, oxidation, ion exchange, and reverse osmosis. [4]. I checked that Ref. 4 "J. Environ. Chem. Eng. 2 (2014) 310‐315". This work is only about sorption of sulfamethoxazole on graphene, but does not include oxidation, ion exchange, reverse osmosis. Please check all and cite the related references. 

3) It is suggested that the Introduction can be rephrased. The background is not clear. The logical structure is not good enough. 

4) Typo errors, such as Line 200, Line 212. 

5) Fig.3. Please index the crystal planes in Fig. 3. 

6) Fig. 5, please add the error bar on all statistical Figures. 

7) Section 3.4.3 is about adsorption thermodynamics analysis, the following reference for this section is required (Xiaoqian Zha et al. Selective Co(II) adsorption using hollow ZIF-8 nanostructures with embedded Fe3O4 nanoparticles.  https://doi.org/10.1021/acsanm.3c05052)

Author Response

(The authors gave the same response as above.)
